# Large Batch Simulation for Deep Reinforcement Learning

**Brennan Shacklett**[1]* **Erik Wijmans**[2] **Aleksei Petrenko**[3,4]
**Manolis Savva**[5] **Dhruv Batra**[2] **Vladlen Koltun**[3] **Kayvon Fatahalian**[1]
[1]Stanford University   [2]Georgia Institute of Technology   [3]Intel Labs
[4]University of Southern California   [5]Simon Fraser University

## Abstract

We accelerate deep reinforcement learning-based training in visually complex 3D environments by two orders of magnitude over prior work, realizing end-to-end training speeds of over 19,000 frames of experience per second on a *single* GPU and up to 72,000 frames per second on a single eight-GPU machine. The key idea of our approach is to design a 3D renderer and embodied navigation simulator around the principle of "batch simulation": accepting and executing large batches of requests simultaneously. Beyond exposing large amounts of work at once, batch simulation allows implementations to amortize in-memory storage of scene assets, rendering work, data loading, and synchronization costs across many simulation requests, dramatically improving the number of simulated agents per GPU and overall simulation throughput. To balance DNN inference and training costs with faster simulation, we also build a computationally efficient policy DNN that maintains high task performance, and modify training algorithms to maintain sample efficiency when training with large mini-batches. By combining batch simulation and DNN performance optimizations, we demonstrate that PointGoal navigation agents can be trained in complex 3D environments on a single GPU in 1.5 days to 97% of the accuracy of agents trained on a prior state-of-the-art system using a 64-GPU cluster over three days. We provide open-source reference implementations of our batch 3D renderer and simulator to facilitate incorporation of these ideas into RL systems.

## 1 Introduction

Speed matters. It is now common for modern reinforcement learning (RL) algorithms leveraging deep neural networks (DNNs) to require *billions* of samples of experience from simulated environments (Wijmans et al., 2020; Petrenko et al., 2020; OpenAI et al., 2019; Silver et al., 2017; Vinyals et al., 2019). For embodied AI tasks such as visual navigation, where the ultimate goal for learned policies is deployment in the real world, learning from *realistic simulations* is important for successful transfer of learned policies to physical robots. In these cases simulators must render detailed 3D scenes and simulate agent interaction with complex environments (Kolve et al., 2017; Dosovitskiy et al., 2017; Savva et al., 2019; Xia et al., 2020; Gan et al., 2020).

Evaluating and training a DNN on billions of simulated samples is computationally expensive. For instance, the DD-PPO system (Wijmans et al., 2020) used 64 GPUs over three days to learn from 2.5 billion frames of experience and achieve near-perfect PointGoal navigation in 3D scanned environments of indoor spaces. At an even larger distributed training scale, OpenAI Five used over 50,000 CPUs and 1000 GPUs to train Dota 2 agents (OpenAI et al., 2019). Unfortunately, experiments at this scale are out of reach for most researchers. This problem will only grow worse as the field explores more complex tasks in more detailed environments.

Many efforts to accelerate deep RL focus on improving the efficiency of DNN evaluation and training – *e.g.*, by "centralizing" computations to facilitate efficient batch execution on GPUs or TPUs (Espeholt et al., 2020; Petrenko et al., 2020) or by parallelizing across GPUs (Wijmans et al., 2020). However, most RL platforms still accelerate environment simulation by running *many copies of off-the-shelf, unmodified simulators*, such as simulators designed for video game engines (Bellemare et al., 2013; Kempka et al., 2016; Beattie et al., 2016; Weihs et al., 2020), on large numbers

---

*Correspondence to bps@cs.stanford.edu

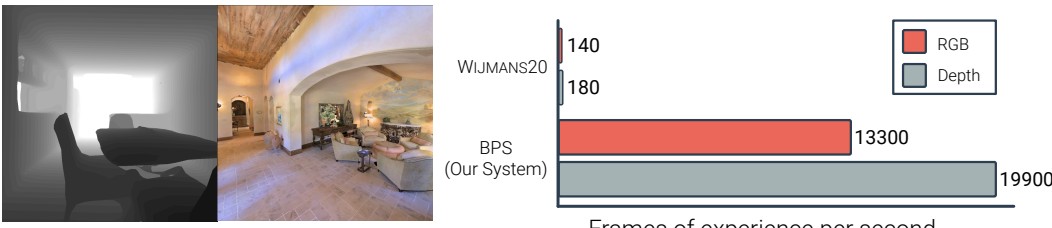

Figure 1: We train agents to perform PointGoal navigation in visually complex Gibson (Xia et al., 2018) and Matterport3D (Chang et al., 2017) environments such as the ones shown here. These environments feature detailed scans of real-world scenes composed of up to 600K triangles and high-resolution textures. Our system is able to train agents using 64×64 depth sensors (a high-resolution example is shown on the left) in these environments at 19,900 frames per second, and agents with 64×64 RGB cameras at 13,300 frames per second on a *single* GPU.

of CPUs or GPUs. This approach is a simple and productive way to improve simulation throughput, but it makes inefficient use of computation resources. For example, when rendering complex environments (Kolve et al., 2017; Savva et al., 2019; Xia et al., 2018), a single simulator instance might consume gigabytes of GPU memory, limiting the total number of instances to far below the parallelism afforded by the machine. Further, running many simulator instances (in particular when they are distributed across machines) can introduce overhead in synchronization and communication with other components of the RL system. Inefficient environment simulation is a major reason RL platforms typically require scale-out parallelism to achieve high end-to-end system throughput.

In this paper, we crack open the simulation black box and take a holistic approach to co-designing a 3D renderer, simulator, and RL training system. Our key contribution is *batch simulation* for RL: designing high-throughput simulators that accept large batches of requests as input (aggregated across different environments, potentially with different assets) and efficiently execute the entire batch at once. Exposing work en masse facilitates a number of optimizations: we reduce memory footprint by sharing scene assets (geometry and textures) across rendering requests (enabling orders of magnitude more environments to be rendered simultaneously on a single GPU), amortize rendering work using GPU commands that draw triangles from multiple scenes at once, hide latency of scene I/O, and exploit batch transfer to reduce data communication and synchronization costs between the simulator, DNN inference, and training. To further improve end-to-end RL speedups, the DNN workload must be optimized to match high simulation throughput, so we design a computationally efficient policy DNN that still achieves high task performance in our experiments. Large-batch simulation increases the number of samples collected per training iteration, so we also employ techniques from large-batch supervised learning to maintain sample efficiency in this regime.

We evaluate batch simulation on the task of PointGoal navigation (Anderson et al., 2018) in 3D scanned Gibson and Matterport3D environments, and show that end-to-end optimization of batched rendering, simulation, inference, and training yields a 110× speedup over state-of-the-art prior systems, while achieving 97% of the task performance for depth-sensor-driven agents and 91% for RGB-camera-driven agents. Concretely, we demonstrate sample generation and training at over 19,000 frames of experience per second on a single GPU.[1] In real-world terms, a single GPU is capable of training a virtual agent on 26 years of experience in a single day.[2] This new performance regime significantly improves the accessibility and efficiency of RL research in realistic 3D environments, and opens new possibilities for more complex embodied tasks in the future.

## 2 RELATED WORK

**Systems for high-performance RL.** Existing systems for high-performance RL have primarily focused on improving the efficiency of DNN components of the workload (policy inference and optimization) and use a simulator designed for efficient *single* agent simulation as a black box. For example, Impala and Ape-X used multiple worker processes to asynchronously collect experience for a centralized learner (Espeholt et al., 2018; Horgan et al., 2018). SEED RL and Sample Factory built upon this idea and introduced inference workers that centralize network inference, thereby allowing it to be accelerated by GPUs or TPUs (Espeholt et al., 2020; Petrenko et al., 2020). DD-PPO proposed a synchronous distributed system for similar purposes (Wijmans et al., 2020). A number

---

[1]Samples of experience used for learning, not 'frameskipped' metrics typically used in Atari/DMLab.

[2]Calculated on rate a physical robot (LoCoBot (Carnegie Mellon University, 2019)) collects observations when operating constantly at maximum speed (0.5 m/s) and capturing 1 frame every 0.25m.

of efficient implementations of these ideas have been proposed as part of RL frameworks or in other deep learning libraries (Liang et al., 2018; Stooke & Abbeel, 2019; Küttler et al., 2019).

We extend the idea of centralizing inference and learning to simulation by cracking open the simulator black box and designing a new simulation architecture for RL workloads. Our large-batch simulator is a drop-in replacement for large numbers of (non-batched) simulation workers, making it synergistic with existing asynchronous and synchronous distributed training schemes. It reduces the number of processes and communication overhead needed for asynchronous methods and eliminates separate simulation worker processes altogether for synchronous methods. We demonstrate this by combining our system with DD-PPO (Wijmans et al., 2020).

Concurrently with our work, CuLE, a GPU-accelerated reimplementation of the Atari Learning Environment (ALE), demonstrates the benefits of centralized batch simulation (Dalton et al., 2020). While both our work and CuLE enable wide-batch execution of their respective simulation workloads, our focus is on high-performance batch rendering of complex 3D environments. This involves optimizations (GPU-driven pipelined geometry culling, 3D asset sharing, and asynchronous data transfer) not addressed by CuLE due to the simplicity of rendering Atari-like environments. Additionally, like CuLE, we observe that the large training batches produced by batch simulation reduce RL sample efficiency. Our work goes further and leverages large-batch optimization techniques from the supervised learning literature to mitigate the loss of sample efficiency without shrinking batch size.

**Large mini-batch optimization.** A consequence of large batch simulation is that more experience is collected between gradient updates. This provides the opportunity to accelerate learning via large mini-batch optimization. In supervised learning, using large mini-batches during optimization typically decreases the generalization performance of models (Keskar et al., 2017). Goyal et al. (2017) demonstrated that model performance can be improved by scaling the learning rate proportionally with the batch size and "warming-up" the learning rate at the start of training. You et al. (2017) proposed an optimizer modification, LARS, that adaptively scales the learning rate at each layer, and applied it to SGD to improve generalization further. In reinforcement learning and natural language processing, the Adam optimizer (Kingma & Ba, 2015) is often used instead of SGD. Lamb (You et al., 2020) combines LARS (You et al., 2017) with Adam (Kingma & Ba, 2015). We do not find that large mini-batch optimization harms generalization in reinforcement learning, but we do find it decreases sample efficiency. We adapt the techniques proposed above – learning rate scaling (You et al., 2017) and the Lamb optimizer (You et al., 2020) – to improve sample efficiency.

**Simulators for machine learning.** Platforms for simulating realistic environments for model training fall into two broad categories: those built on top of pre-existing game engines (Kolve et al., 2017; Dosovitskiy et al., 2017; Lee et al., 2019; Gan et al., 2020; James et al., 2020), and those built from scratch using open-source 3D graphics and physics libraries (Savva et al., 2017; 2019; Xia et al., 2018; 2020; Xiang et al., 2020; Zeng et al., 2020). While improving simulator performance has been a focus of this line of work, it has been evaluated in a narrow sense (*i.e.* frame rate benchmarks for predetermined agent trajectories), not accounting for the overall performance of end-to-end RL training. We instead take a holistic approach to co-design rendering and simulation modules and their interfaces to the RL training system, obtaining significant gains in end-to-end throughput over the state of the art.

## 3 SYSTEM DESIGN & IMPLEMENTATION

Batch simulation accelerates rollout generation during RL training by processing many simulated environments simultaneously in large batches. Fig. 2 illustrates how batch simulation interacts with policy inference to generate rollouts. Simulation for sensorymotor agents, such as the PointGoal navigation task targeted by our implementation, can be separated into two tasks: determining the next environment state given an agent's actions and rendering its sensory observations. Therefore, our design utilizes two components: a batch simulator that performs geodesic distance and navigation mesh (Snook, 2000) computations on the CPU, and a batch renderer that renders complex 3D environments on the GPU.

During rollout generation, batches of requests are passed between these components – given $N$ agents, the simulator produces a batch of $N$ environment states. Next, the renderer processes the batch of environment states by simultaneously rendering $N$ frames and exposing the result directly in GPU memory. Agent observations (from both the simulator and the renderer) are then provided as a batch to policy inference to determine the next actions for the $N$ agents.

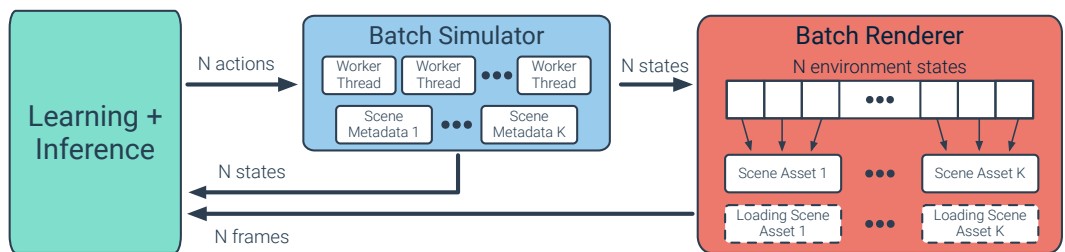

Figure 2: The batch simulation and rendering architecture. Each component communicates at the granularity of batches of $N$ elements (e.g., $N$=1024), minimizing communication overheads and allowing components to independently parallelize their execution over each batch. To fit the working set for large batches on the GPU, the renderer maintains $K \ll N$ unique scene assets in GPU memory and shares these assets across subsets of the $N$ environments in a batch. To enable experience collection across a diverse set of environments, the renderer continuously updates the set of $K$ in-memory scene assets using asynchronous transfers that overlap rollout generation and learning.

The key idea is that the batch simulator and renderer implementations (in addition to the DNN workload) *take responsibility for their own parallelization*. Large batch sizes (values of $N$ on the order of hundreds to thousands of environments) provide opportunities for implementations to efficiently utilize parallel execution resources (e.g., GPUs) as well as amortize processing, synchronization, and data communication costs across many environments. The remainder of this section describes the design and key implementation details of our system's batch simulator and batch renderer, as well as contributions that improve the efficiency of policy inference and optimization in this regime.

### 3.1 BATCH ENVIRONMENT SIMULATION

Our CPU-based batch simulator executes geodesic distance and navigation mesh computations in parallel for a large batch of environments. Due to differences in navigation mesh complexity across environments, the time to perform simulation may differ per environment. This variance is the source of workload imbalance problems in parallel synchronous RL systems (Wijmans et al., 2020; Savva et al., 2019) and one motivation for recent asynchronous designs (Petrenko et al., 2020; Espeholt et al., 2020; 2018). To ensure good workload balance, our batch simulator operates on large batches that contain *significantly more* environments than the number of available CPU cores and dynamically schedules work onto cores using a pool of worker threads (simulation for each environment is carried out sequentially). Worker threads report simulation results into a designated per-environment slot in a results buffer that is communicated to the renderer via a single batched request when all environment simulation for a batch is complete. To minimize CPU memory usage, the simulator only loads navigation meshes and does not utilize the main rendering assets.

### 3.2 BATCH RENDERING

A renderer for producing RL agent observations in scanned real-world environments must efficiently synthesize many low-resolution renderings (e.g., 64×64 pixels) of scenes featuring high-resolution textures and complex meshes.[3] Low-resolution output presents challenges for GPU acceleration. Rendering images one at a time produces too little rendering work to efficiently utilize a modern GPU rendering pipeline's parallel processing resources. Rendering many environments concurrently but individually (e.g., from different worker threads or processes) exposes more rendering work to the GPU, but incurs the overhead of sending the GPU many fine-grained rendering commands.

To address the problem of rendering many small images efficiently, our renderer combines the GPU commands required to render observations for an entire simulation batch of $N$ environments into a *single rendering request* to the GPU – effectively drawing the entire batch as a single large frame (individual environment observations are tiles in the image). This approach exposes large amounts of rendering work to the GPU and amortizes GPU pipeline configuration and rendering overhead over an entire batch. Our implementation makes use of modern GPU pipeline features (Khronos Group, 2017) that allow rendering tasks that access different texture and mesh assets to proceed as part of a single large operation (avoiding GPU pipeline flushes due to pipeline state reconfiguration).

**Scene asset sharing.** Efficiently utilizing a GPU requires batches to be large (we use $N$ up to 1024). However, geometry and texture assets for a single environment may be gigabytes in size, so naively loading unique assets for each environment in a large batch would exceed available GPU memory. Our implementation allows multiple environments in a batch to reference the same 3D scene assets in

---

[3]The Matterport3D dataset contains up to 600K triangles per 3D scan.

GPU memory. Specifically, our system materializes $K$ unique assets in GPU memory ($K \ll N$) and constructs batches of $N$ environments that reference these assets. Asset reuse decreases the diversity of training experiences in a batch, so to preserve diversity we limit the ratio of $N$ to $K$ in any one batch to 32, and continuously rotate the set of $K$ assets in GPU memory. The renderer refreshes the set of $K$ assets by asynchronously loading new scene assets into GPU memory during the main rollout generation and learning loop. As episodes complete, new environments are constructed to reference the newly loaded assets, and assets no longer referenced by active environments are removed from GPU memory. This design allows policy optimization to learn from an entire dataset of assets without exceeding GPU memory or incurring the latency costs of frequent asset loading.

**Pipelined geometry culling.** When rendering detailed geometry to low-resolution images, most scene triangles cover less than one pixel. As a result, rendering performance is determined by the rate the GPU's rasterization hardware processes triangles, not the rate the GPU can shade covered pixels. To reduce the number of triangles the GPU pipeline must process, the renderer uses idle GPU cores to identify and discard geometry that lies outside the agent's view—a process known as frustum culling (Akenine-Möller et al., 2018). Our implementation pipelines frustum culling operations (implemented using GPU compute shaders) with rendering for different environments in a batch. This pipelined design increases GPU utilization by concurrently executing culling work on the GPU's programmable cores and rendering work on the GPU's rasterization hardware.

### 3.3 POLICY DNN ARCHITECTURE

High-throughput batch simulation creates a need for high-throughput policy DNN inference. Therefore, we develop a policy DNN architecture designed to achieve an efficient balance between high task performance and low computational cost. Prior work in PointGoal navigation (Wijmans et al., 2020) used a policy DNN design where a visual encoder CNN processes an agent's visual sensory information followed by an LSTM (Hochreiter & Schmidhuber, 1997) that determines the policy's actions. Our policy DNN uses this core design augmented with several performance optimizations.

First, we reduce DNN effective input resolution from $128{\times}128$ (Wijmans et al., 2020) to $64{\times}64$. Beyond this simple optimization, we choose a shallow visual encoder CNN – a nine-layer ResNet (He et al., 2016) (ResNet18 with every other block removed), rather than the 50 layer (or larger) ResNets used by prior work. To counteract reduced task performance from the ResNet's relatively low capacity, all stages include Squeeze-Excite (SE) blocks (Hu et al., 2018) with $r{=}16$. Additionally, we use a `SpaceToDepth` stem (Ridnik et al., 2020), which we find performs equally to the standard `Conv+MaxPool` stem while using less GPU memory and compute.

Finally, we avoid the use of normalization layers in the ResNet as these require spatial reductions over the feature maps, preventing layer-fusion optimizations. Instead, the CNN utilizes Fixup Initialization (Zhang et al., 2019) to improve training stability. Fixup Initialization replaces expensive normalization layers after each convolution with cheap elementwise multiplication and addition.

### 3.4 LARGE MINI-BATCH POLICY OPTIMIZATION

In on-policy reinforcement learning, policy optimization utilizes trajectories of experience to reduce bias and for backpropagation-through-time. When generating trajectories of length $L$ with a simulation batch size of $N$, a rollout will have $N{\times}L$ steps of experience. Therefore, a consequence of simulation with large $N$ is that more experience is collected per rollout.

Large $N$ presents the opportunity to utilize large mini-batches to improve the throughput of policy optimization; however, throughput must be balanced against generalization and sample efficiency to ensure that reduced task performance does not offset the throughput gains. Although large mini-batch training is known to hurt generalization in supervised learning (Keskar et al., 2017), we do not see evidence of this for RL. Conversely, we do find that sample efficiency for PointGoal navigation is harmed by naively increasing $N$. Fortunately, we are able to mitigate this loss of sample efficiency using techniques for improving generalization from the large mini-batch optimization literature.

First, we scale the learning rate by $\sqrt{\frac{B}{B_{\text{base}}}}$, where $B_{\text{base}}{=}256$ and $B$, the training batch size, is $N{\times}L$ divided by the number of mini-batches per training iteration. We find it beneficial to use the scaled learning rate immediately instead of 'warming-up' the learning rate (Goyal et al., 2017). Second, we use and adapt the Lamb optimizer (You et al., 2020). Lamb is a modification to Adam (Kingma & Ba, 2015) that applies LARS (You et al., 2017) to the step direction estimated by Adam to better handle high learning rates. Since the Adam optimizer is often used with PPO (Schulman et al.,

2017), Lamb is a natural choice. Given the Adam step direction $s_t^{(k)}$ for weights $\theta_t^{(k)}$,

$$\theta_{t+1}^{(k)} = \theta_t^{(k)} - \eta_t r_t^{(k)} (s_t^{(k)} + \lambda \theta_t^{(k)}) \qquad\qquad r_t^{(k)} = \frac{\phi(||\theta_t^{(k)}||)}{||s_t^{(k)} + \lambda \theta_t^{(k)}||} \qquad (1)$$

where $\eta_t$ is the learning rate and $\lambda$ is the weight decay coefficient. We set $\phi(||\theta_t^{(k)}||)$ as $\min\{||\theta_t^{(k)}||, 10.0\}$ and introduce an additional clip on the trust ratio $r_t^{(k)}$:

$$r_t^{(k)} = \min\left\{ \max\left\{ \frac{\phi(||\theta_t^{(k)}||)}{||s_t^{(k)} + \lambda \theta_t^{(k)}||}, \rho \right\}, \frac{1}{\rho} \right\} \qquad (2)$$

We find the exact value of $\rho$ to be flexible (we observed similar training with $\rho \in \{10^{-2}, 10^{-3}, 10^{-4}\}$) and also observed that this clip is only influential at the start of training, suggesting that there is an initialization scheme where it is unnecessary.

## 4  RESULTS

We evaluate the impact of our contributions on end-to-end training speed and task performance by training PointGoal navigation agents in the complex Gibson (Xia et al., 2018) and Matterport3D (Chang et al., 2017) environments. The fastest published end-to-end training performance in these environments is achieved with the synchronous RL implementation presented with DD-PPO (Wijmans et al., 2020). Therefore, both our implementation and the baselines we compare against are synchronous PPO-based RL systems.

### 4.1  EXPERIMENTAL SETUP

**PointGoal navigation task.** We train and evaluate agents via the same procedure as Wijmans et al. (2020): agents are trained for `PointGoalNav` (Anderson et al., 2018) with either a `Depth` sensor or an RGB camera. `Depth` agents are trained on Gibson-2plus (Xia et al., 2018) and, consistent with Wijmans et al. (2020), RGB agents are also trained on Matterport3D (Chang et al., 2017). RGB camera simulation requires textures for the renderer, increasing the GPU memory consumed by each scene significantly. Both classes of agent are trained on 2.5 billion simulated samples of experience.

Agents are evaluated on the Gibson dataset (Xia et al., 2018). We use two metrics: Success, whether or not the agent reached the goal, and SPL (Anderson et al., 2018), a measure of both Success and efficiency of the agent's path. We perform policy evaluation using Habitat-Sim (Savva et al., 2019), unmodified for direct comparability to prior work.

**Batch Processing Simulator (BPS).** We provide an RL system for learning `PointGoalNav` built around the batch simulation techniques and system-wide optimizations described in Section 3. The remainder of the paper refers to this system as BPS (Batch Processing Simulator). To further accelerate the policy DNN workload, BPS uses half-precision inference and mixed-precision training.

**Baseline.** The primary baseline for this work is Wijmans et al. (2020)'s open-source `PointGoalNav` implementation, which uses Habitat-Sim (Savva et al., 2019) – the prior state of the art in high-performance simulation of realistic environments such as Gibson. Unlike BPS, multiple environments are simulated simultaneously using parallel worker processes that render frames at $256\times256$ pixels before downsampling to $128\times128$ for the visual encoder. The fastest published configuration uses a ResNet50 visual encoder. Subsequent sections refer to this implementation as WIJMANS20.

**Ablations.** As an additional baseline, we provide WIJMANS++, which uses the optimized SE-ResNet9-based policy DNN (including performance optimizations and resolution reduction relative to WIJMANS20) developed for BPS, but otherwise uses the same system design and simulator as WIJMANS20 (with a minor modification to not load textures for `Depth` agents). WIJMANS++ serves to isolate the impact of two components of BPS: first, the low-level DNN efficiency improvements, and, more importantly, the performance of batch simulation versus WIJMANS20's independent simulation worker design. Additionally, to ablate the effect of our encoder CNN architecture optimizations, we include a variant of BPS, BPS-R50, that uses the same ResNet50 visual encoder and input resolution as WIJMANS20, while maintaining the other of optimizations BPS.

**Multi-GPU training.** To support multi-GPU training, all three systems replace standard PPO with DD-PPO (Wijmans et al., 2020). DD-PPO scales rollout generation and policy optimization across all available GPUs, scaling the number of environments simulated and the number of samples gathered between training iterations proportionally. We report results with eight GPUs.

| Sensor | System | CNN | Agent Res. | RTX 3090 | RTX 2080Ti | Tesla V100 | 8×2080Ti | 8×V100 |
|--------|--------|-----|------------|----------|------------|------------|----------|--------|
| Depth | BPS | SE-ResNet9 | 64 | 19900 | 12900 | 12600 | 72000 | 46900 |
| | BPS-R50 | ResNet50 | 128 | 2300 | 1400 | 2500 | 10800 | 18400 |
| | WIJMANS++ | SE-ResNet9 | 64 | 2800 | 2800 | 2100 | 9300 | 13100 |
| | WIJMANS20 | ResNet50 | 128 | 180 | 230 | 200 | 1600 | 1360 |
| RGB | BPS | SE-ResNet9 | 64 | 13300 | 8400 | 9000 | 43000 | 37800 |
| | BPS-R50 | ResNet50 | 128 | 2000 | 1050 | 2200 | 6800 | 14300 |
| | WIJMANS++ | SE-ResNet9 | 64 | 990 | 860 | 1500 | 4600 | 8400 |
| | WIJMANS20 | ResNet50 | 128 | 140 | OOM | 190 | OOM | 1320 |

Table 1: **System performance.** Average frames per second (FPS, measured as samples of experience processed per second) achieved by each system. BPS achieves a speedup of $110\times$ over WIJMANS20 on Depth experiments (19,900 vs. 180 FPS) and $95\times$ on RGB experiments (13,300 vs. 140 FPS) on an RTX 3090 GPU. OOM (out of memory) indicates that the RTX 2080Ti could not run WIJMANS20 with the published DD-PPO system parameters due to insufficient GPU memory.

| | Sensor | System | Validation | | Test | |
|---|--------|--------|------------|----------|------|---------|
| | | | SPL | Success | SPL | Success |
| 1 | Depth | BPS | $94.4_{\pm0.7}$ | $99.2_{\pm1.4}$ | 91.5 | 97.3 |
| 2 | | WIJMANS20 | $95.6_{\pm0.3}$ | $99.9_{\pm0.2}$ | 94.4 | 98.2 |
| 3 | RGB | BPS | $88.4_{[\pm0.9}$ | $97.6_{\pm0.3}$ | 83.7 | 95.7 |
| 4 | | BPS @ 128×128 | $87.8_{\pm0.7}$ | $97.3_{\pm0.4}$ | 85.6 | 96.3 |
| 5 | | WIJMANS20 | 92.9 | 99.1 | 92.0 | 97.7 |

Table 2: **Policy performance.** SPL and Success of agents produced by BPS and WIJMANS20. The performance of the BPS agent is within the margin of error of the WIJMANS20 agent for Depth experiments on the validation set, and within five percent on RGB. BPS agents are trained on eight GPUs with aggregate batch size $N$=1024.

**Determining batch size.** The per-GPU batch size, $N$, controls a trade-off between memory usage, sample efficiency, and speed. For BPS, $N$ designates the batch size for simulation, inference, and training. For WIJMANS20 and WIJMANS++, $N$ designates the batch size for inference and training, as well as the number of simulation processes. WIJMANS20 sets $N$=4 for consistency with Wijmans et al. (2020). To maximize performance of single-GPU runs, BPS uses the largest batch size that fits in GPU memory, subject to the constraint that no one scene asset can be shared by more than 32 environments in the batch. In eight-GPU configurations, DD-PPO scales the number of parallel rollouts with the number of GPUs, so to maintain reasonable sample efficiency BPS limits per-GPU batch size to $N$=128, with $K$=4 active scenes per GPU. WIJMANS++ Depth experiments use $N$=64 (limited by system memory due to $N$ separate processes running Habitat-Sim). Batch size in WIJMANS++ RGB experiments is limited by GPU memory ($N$ ranges from 6 to 20 depending on the GPU). Appendix B provides the batch sizes used in all experiments.

**Benchmark evaluation.** We report end-to-end performance benchmarks in terms of average frames per second (FPS) achieved by each system. We measure FPS as the number of samples of experience processed over 16,000 inference batches divided by the time to complete rollout generation and training for those samples. In experiments that run at 128×128 pixel sensor resolution, rendering occurs at 256×256 and is downsampled for the policy DNN to match the behavior of WIJMANS20 regardless of system, while 64×64 resolution experiments render without downsampling. Results are reported across three models of NVIDIA GPUs: Tesla V100, GeForce RTX 2080Ti, and GeForce RTX 3090. (The different GPUs are also accompanied by different CPUs, see Appendix C.)

### 4.2 END-TO-END TRAINING SPEED

**Single-GPU performance.** On a single GPU, BPS trains agents $45\times$ (9000 vs. 190 FPS, Tesla V100) to $110\times$ (19900 vs. 180 FPS, RTX 3090) faster than WIJMANS20 (Table 1). The greatest speedup was achieved using the RTX 3090, which trains Depth agents at 19,900 FPS and RGB agents at 13,300 FPS – a $110\times$ and $95\times$ increase over WIJMANS20, respectively. This 6000 FPS performance drop from Depth to RGB is not caused by the more complex rendering workload, because the addi-

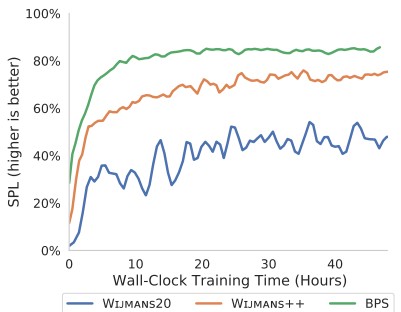

Figure 3: SPL vs. wall-clock time (RGB agents) on a RTX 3090 over 48 hours (time required to reach 2.5 billion samples with BPS). BPS exceeds $80\%$ SPL in 10 hours and achieves a significantly higher SPL than the baselines.

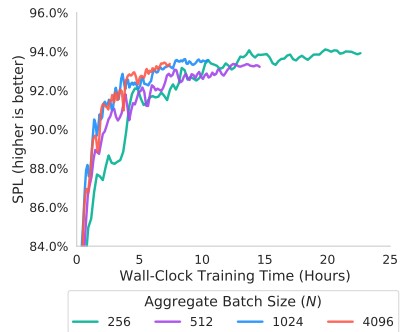

Figure 4: SPL vs. wall-clock time (BPS training Depth agents over 2.5 billion samples on 8 Tesla V100s) for various batch sizes ($N$). $N$=256 finishes after $2\times$ the wall-clock time as $N$=1024, but both achieve statistically similar SPL.

tional cost of fetching RGB textures is masked by the dominant cost of geometry processing. Instead, due to memory constraints, BPS must reduce the batch size ($N$) for RGB tasks, reducing the performance of all components (further detail in Section 4.4).

To assess how much of the BPS speedup is due to the SE-ResNet9 visual encoder and lower input resolution, we also compare BPS-R50 and WIJMANS20, which have matching encoder architecture and resolution. For Depth agents training on the the RTX 3090, BPS-R50 still achieves greater than $10\times$ performance improvement over WIJMANS20 (2,300 vs. 180 FPS), demonstrating the benefits of batch simulation even in DNN heavy workloads. BPS-R50 is only $6\times$ faster than WIJMANS20 on the RTX 2080Ti, since the ResNet50 encoder's larger memory footprint requires batch size to be reduced from $N$=128 on the RTX 3090 (24 GB RAM) to $N$=64 on the RTX 2080Ti (11 GB RAM). Similarly, increasing DNN input resolution increases memory usage, forcing batch size to be decreased and reducing performance (Table A1).

The BPS batch simulation architecture is significantly faster than the WIJMANS++ design that uses multiple worker processes. When training Depth agents, BPS outperforms WIJMANS++ by $4.5\times$ to $7.8\times$, with a greater speedup of $6\times$ to $13\times$ for RGB agents. Since BPS and WIJMANS++ use the same policy DNN and input resolution, this comparison isolates the performance advantage of batch simulation and rendering against an optimized version of the multiple-worker-process-based design: WIJMANS++ is up to $15\times$ faster than WIJMANS20. The relative speedup of BPS for RGB agents is larger because WIJMANS++ does not share environment assets between simulator instances. Textures needed for RGB rendering significantly increase the memory footprint of each simulator instance and limit WIJMANS++ to as few as $N$=6 workers (compared to $N$=64 for Depth agents). Conversely, BPS shares 3D assets across environments and maintains a batch size at least $N$=128 for RGB agents.

**Multi-GPU performance.** BPS achieves high end-to-end throughput when running in eight-GPU configurations: up to 72,000 FPS for Depth agents on eight RTX 2080Ti. Relative to WIJMANS20, BPS is $29\times$ to $34\times$ faster with eight Telsa V100s and $45\times$ faster with eight RTX 2080Ti. These speedups are lower than the single-GPU configurations, because BPS reduces the per-GPU batch size in eight-GPU configurations to avoid large aggregate batches that harm sample efficiency. This leads to imperfect multi-GPU scaling for BPS: for Depth agents, each RTX 2080Ti is approximately 4000 FPS slower in an eight-GPU configuration than in a single-GPU configuration. Eight-GPU scaling for Depth is lower on the Tesla V100s ($3.7\times$) compared to the 2080Ti ($5.6\times$) because larger batch sizes are needed to utilize the large number of parallel compute units on the Tesla V100.

### 4.3 POLICY TASK PERFORMANCE

To understand how the system design and visual encoder architecture of BPS impact learning, we evaluate the task performance of agents trained with BPS in an eight-GPU configuration with aggregate batch size of $N$=1024. For Depth agents, the reduction in encoder CNN depth results in a $1\%$ and $3\%$ decrease in SPL on Val and Test respectively with a negligible Success change on Val and a 0.9 Success decrease on Test (Table 2, row 1 vs. 2). For RGB agents, BPS suffers a performance loss of 3.8/1.3 SPL/Success on Val and 8.3/2.0 SPL/Success on Test (Table 2, row 3 vs. 4). Despite this performance reduction, the RGB agent trained by BPS would have won the 2019 Habitat challenge by 4 SPL and is only beaten by WIJMANS20's ResNet50-based policy on Test.

**SPL vs. training time.** BPS significantly outperforms the baselines in terms of wall-clock training time to reach a given SPL. After 10 hours of training on a single RTX 3090, BPS reaches over $80\%$ SPL (on Val) while WIJMANS20 and WIJMANS++ reach only $40\%$ and $65\%$ SPL respectively (Fig. 3). Furthermore, BPS converges within $1\%$ of peak SPL at approximately 20 hours; conversely, neither baseline reaches convergence within 48 hours. BPS converges to a lower final SPL in Fig. 3 than Table 2, likely due to the tested single-GPU configuration differing in batch size and scene asset swapping frequency compared to the eight-GPU configuration used to produce Table 2.

**Effect of batch size.** The end-to-end training efficiency of BPS is dependent on batch size ($N$): larger $N$ will increase throughput and reduce wall-clock time to reach a given number of samples, but may harm sample efficiency and final task performance at convergence. We evaluate this relationship by training Depth agents with BPS across a range of $N$. As shown in Fig. 4, all experiments converge within $1\%$ of the peak SPL achieved; however, $N$=256 halves total throughput compared to $N$=1024 (the setting used elsewhere in the paper for eight-GPU configurations). At the high end, $N$=4096 yields slightly worse SPL than $N$=1024 and is only 20% faster. Larger batch sizes also require more memory for rollout storage and training, which is prohibitive for RGB experiments that require significant GPU memory for texture assets. In terms of sample efficiency alone, Fig. A1 shows that smaller batch sizes have a slight advantage (without considering training speed).

## 4.4 RUNTIME BREAKDOWN

Fig. 5 provides a breakdown of time spent in each of the main components of the BPS system ($\mu$s per frame). Nearly 60% of BPS runtime on the RTX 3090 GPU (for both Depth and RGB) is spent in DNN inference and training, even when rendering complex 3D environments and using a small, low-cost policy DNN. This demonstrates the high degree of simulation efficiency achieved by BPS. Furthermore, the results in Table A2 for BPS-R50 show that, with the larger visual encoder, over 90% of per-frame time (on Depth tasks) is spent in the DNN workload (70% on learning).

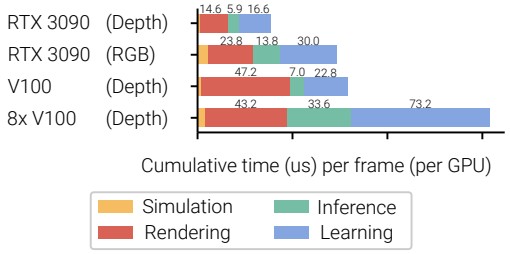

Figure 5: BPS runtime breakdown. Inference represents policy evaluation cost during rollout generation. Learning represents the total cost of policy optimization.

Batch size ($N$) heavily impacts DNN performance. DNN operations for Depth ($N$=1024) are 2× faster than RGB ($N$=256) on the RTX 3090, because RGB must use a smaller batch size to fit texture assets in GPU memory. The larger batch size improves GPU utilization for all system components. A similar effect is visible when comparing the single-GPU and eight-GPU V100 breakdowns. BPS reduces the per-GPU batch size from $N$=1024 to $N$=128 in eight-GPU experiments to maintain an aggregate batch size of 1024 for sample efficiency. Further work in policy optimization to address this learning limitation would improve multi-GPU scaling by allowing larger aggregate batch sizes.

## 5 DISCUSSION

We demonstrated that architecting an RL training system around the idea of batch simulation can accelerate learning in complex 3D environments by one to two orders of magnitude over prior work. With these efficiency gains, agents can be trained with billions of simulated samples from complex environments in about a day using only a single GPU. We believe these fast turnaround times stand to make RL in realistic simulated environments accessible to a broad range of researchers, increase the scale and complexity of tasks and environments that can be explored, and facilitate new studies of how much visual realism is needed to learn a given task (e.g., dynamic lighting, shadows, custom augmentations). To facilitate such efforts, our system is available open-source at https://github.com/shacklettbp/bps-nav.

More generally, this work demonstrates the value of building RL systems around components that have been specifically designed for RL workloads, not repurposed from other application domains. We believe this philosophy should be applied to other components of future RL systems, in particular to new systems for performing physics simulation in complex environments.

ACKNOWLEDGMENTS

This work was supported in part by NSF, DARPA, ONR YIP, ARO PECASE, Intel, and Facebook. EW is supported in part by an ARCS fellowship. We thank NVIDIA for GPU equipment donations. We also thank the Habitat team for helpful discussions and their support of this project.

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

| Sensor | System | Agent Res. | RTX 3090 | RTX 2080Ti | Tesla V100 | 8×2080Ti | 8×V100 |
|---|---|---|---|---|---|---|---|
| Depth | BPS | 64 | 19900 | 12900 | 12600 | 72000 | 46900 |
| | BPS | 128 | 6900 | 4880 | 5800 | 38000 | 41100 |
| | BPS-R50 | 64 | 4800 | 2700 | 4000 | 19400 | 26500 |
| | BPS-R50 | 128 | 2300 | 1400 | 2500 | 10800 | 18400 |
| RGB | BPS | 64 | 13300 | 8400 | 9000 | 43000 | 37800 |
| | BPS | 128 | 6100 | 3600 | 4800 | 22300 | 31100 |
| | BPS-R50 | 64 | 4000 | 2100 | 3500 | 14100 | 19700 |
| | BPS-R50 | 128 | 2000 | 1050 | 2200 | 6800 | 14300 |

Table A1: **Impact of Visual Encoder Input Resolution on Performance.** Resolution has the largest impact on performance when increased memory usage forces BPS' batch size to be decreased. For example, on a single Tesla V100, BPS' Depth performance drops by $2.2\times$ after increasing the resolution, because batch size decreases from $N{=}1024$ to $N{=}512$. Conversely, the eight-GPU Tesla V100 results only show a 12% decrease in performance, since batch size is fixed at $N{=}128$. Experiments with $128{\times}128$ pixel resolution are rendered at $256{\times}256$ and downsampled.

| Sensor | System | CNN | Simulation + Rendering | Inference | Learning |
|---|---|---|---|---|---|
| Depth | BPS | SE-ResNet9 | 16.1 | 5.9 | 16.6 |
| | BPS-R50 | ResNet50 | 26.9 | 99.3 | 311.3 |
| | WIJMANS++ | SE-ResNet9 | 270.9 | 78.8 | 42.8 |
| | WIJMANS20 | ResNet50 | 1901.3 | 3968.6 | 1534.5 |
| RGB | BPS | SE-ResNet9 | 29.6 | 13.8 | 30.0 |
| | BPS-R50 | ResNet50 | 40.3 | 110.2 | 333.4 |
| | WIJMANS++ | SE-ResNet9 | 520.3 | 389.5 | 169.3 |
| | WIJMANS20 | ResNet50 | 1911.1 | 4027.5 | 1587.5 |

Table A2: **Runtime breakdown across systems.** Microseconds per frame for each RL component on a RTX 3090. SE-ResNet9 uses an input resolution of 64x64, while ResNet50 uses an input resolution of 128x128. Note the large amount of time spent by WIJMANS20 on policy inference, caused by GPU memory constraints that force a small number of rollouts per iteration. BPS-R50's performance is dominated by the DNN workload due to the large ResNet50 visual encoder.

## A  ADDITIONAL RESULTS

### A.1  FLEE AND EXPLORE TASKS ON AI2-THOR DATASET

To demonstrate batch simulation and rendering on additional tasks besides PointGoal navigation, BPS also supports the Flee (find the farthest valid location from a given point) and Explore (visit as much of an area as possible) tasks. We evaluate BPS's performance on these tasks on the AI2-THOR (Kolve et al., 2017) dataset to additionally show how batch rendering performs on assets with less geometric complexity than the scanned geometry in Gibson and Matterport3D.

Table A3 shows the learned task performance and end-to-end training speed of BPS on these two tasks for Depth-sensor-driven agents. For both tasks, BPS outperforms its results on PointGoal navigation by around 5000 frames per second, largely due to the significantly reduced geometric complexity of the AI2-THOR dataset versus Gibson. Additionally, the Explore task slightly outperforms the Flee task by 600 FPS on average due to a simpler simulation workload, because no geodesic distance computation is necessary.

### A.2  STANDALONE BATCH RENDERER PERFORMANCE

To evaluate the absolute performance of BPS's batch renderer independently from other components of the system, Fig. A2 shows the performance of the standalone renderer on the "Stokes" scene from the Gibson dataset using a set of camera positions taken from a training run. A batch size of 512

| Task | FPS | Training Score | Validation Score |
|---|---|---|---|
| Explore | 25300 | 6.42 | 5.61 |
| Flee | 24700 | 4.27 | 3.65 |

Table A3: **Task and FPS results for Flee and Explore tasks** with `Depth` agents (on a RTX 3090), where the Training / Validation Score is measured in meters for the Flee task and number of cells visited on the navigation mesh for the Explore task. These tasks achieve higher throughput than PointGoal navigation due to the lower complexity AI2-THOR meshes used. The relatively low scores are a result of the small spatial size of the AI2-THOR assets.

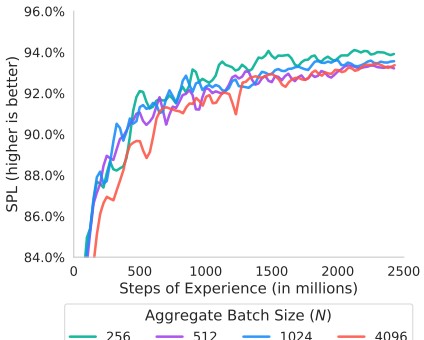

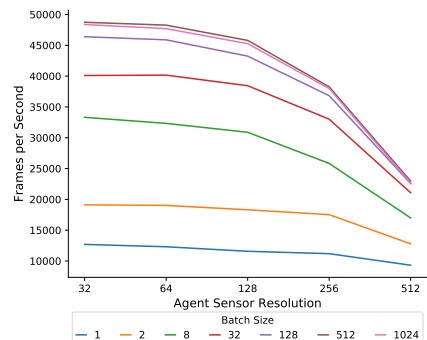

Figure A1: BPS's validation set SPL for `Depth` vs. number of training samples across a range of batch sizes. This graph shows that sample efficiency slightly decreases with larger batch sizes (with the exception of $N$=512 vs. $N$=1024, where $N$=1024 exhibits better validation score). Ultimately, the difference in converged performance is less than 1% SPL between different batch sizes. Although $N$=256 converges the fastest in terms of training samples needed, Fig. 4 shows that $N$=256 performs poorly in terms of SPL achieved per unit of training time.

Figure A2: Frames per second achieved by the standalone renderer on a RTX 3090 across a range of resolutions and batch sizes for a RGB sensor on the Gibson dataset. Performance saturates at a batch size of 512. For lower batch sizes, increasing resolution has a minimal performance impact, because the GPU still isn't fully utilized. As resolution increases with larger batches, the relative decrease in performance from higher resolution increases.

achieves a 3.7x performance increase over a batch size of 1, which emphasizes the fact that much of the end to end speedup provided by batch rendering comes from the performance benefits of larger inference and training batches made possible by the batch renderer's 3D asset sharing.

Fig. A2 also demonstrates that the batch renderer can maintain extremely high performance (approximately 23,000 FPS) at much higher resolutions than used in the RL tasks presented in this work. While this may be useful for tasks requiring higher resolution inputs, considerable advancements would need to be made in DNN performance to handle these high resolution frames at a comparable framerate to the renderer.

## A.3   LAMB OPTIMIZER ABLATION STUDY

To demonstrate the benefit provided by the Lamb optimizer with regard to sample efficiency, Fig. A3 shows a comparison between the Lamb optimizer used by BPS and the Adam optimizer used by WIJMANS20 and WIJMANS++. The training setup for these two optimizers is identical, with the exception of the removal of learning rate scaling for Adam, as this causes training to diverge. The benefits of Lamb are most pronounced early in training, allowing Lamb to reach within 0.7% SPL of convergence after just 1 billion samples of experience (while Adam trails Lamb by 1.5% at the same point). As training progresses, the difference shrinks as Adam slowly converges for a final difference of 0.6% SPL after 2.5 billion frames.

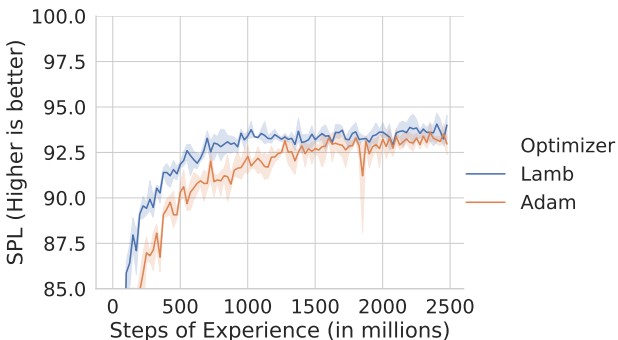

Figure A3: The effect of the Lamb optimizer versus the baseline Adam optimizer on sample efficiency while training a `Depth` sensor driven agent. Lamb maintains a consistent lead in terms of SPL throughout training, especially in the first half of training.

## B  EXPERIMENT AND TRAINING ADDITIONAL DETAILS

**Complete PointGoal navigation description.** We train and evaluate agents via the same procedure as Wijmans et al. (2020). Specifically, agents are trained for `PointGoalNav` (Anderson et al., 2018) where the agent is tasked with navigating to a point specified relative to its initial location. Agents are equipped with a GPS+Compass sensor (providing the agent with its position and orientation relative to the starting position) and either a `Depth` sensor or `RGB` camera. The agent has access to 4 low-level actions, `forward` (0.25m), `turn_left`(10°), `turn_right`(10°), and `stop`.

Agents are evaluated on the Gibson dataset (Xia et al., 2018). We use two metrics to evaluate the agents: Success, whether or not the agent called `stop` within 0.2m of the goal, and SPL (Anderson et al., 2018), a measure of both Success and efficiency of the agent's path. During evaluation, the agent does not have access to reward.

**Half-precision inference and mixed-precision training.** We perform inference in half precision for all components except the action distribution. We train in mixed precision (Jia et al., 2018), utilizing the Apex library in O2 mode. We use half precision for all computations except the action distribution and losses. Additionally, The optimizer still utilizes single precision for all computations and applies gradients to a single-precision copy of the weights.

**Training hyper-parameters** Our hyper-parameters for eight-GPU runs are given in Table A4. We additionally employ a gradual learning rate decay where we decay the learning rate from its scaled value back to the base value over the first half of training. We use a cosine schedule.

We find it necessary to set $\rho$=1.0 for the bias parameters, fixup parameters, and layer-norm parameters of the network, making the optimizer for these parameters equivalent to AdamW (Kingma & Ba, 2015; Loshchilov & Hutter, 2018). We also use L2 weight-decay both to add back regularization lost by removing normalization layers and to stabilize Lamb; we use $\lambda$=$10^{-2}$.

We find one epoch of PPO with two mini-batches to be sufficient (instead of two epochs with two mini-batches), thus effectively doubling the learning speed. We also evaluated one mini-batch, but found two to be beneficial while also having little penalty on overall training speed.

## C  BENCHMARKING ADDITIONAL DETAILS

**Pretrained benchmarking.** A pretrained DNN is used when benchmarking to avoid frequent environment resets at the start of training.

**Benchmarking hyper-parameters.** Table A5 shows the setting for hyper-parameters that impact system throughput.

**GPU details** We report FPS results on three models of NVIDIA GPUs: Tesla V100, GeForce RTX 2080 TI, and GeForce RTX 3090. We demonstrate scaling to multiple GPUs with eight GPU configurations for all but the RTX 3090. Single GPU and eight GPU results are benchmarked on the same machines; however single GPU configurations are limited to 12 cores and 64 GB of RAM as this is a reasonable configuration for a single GPU workstation.

| PPO Parameters | |
|---|---|
| PPO Epochs | 1 |
| PPO Mini-Batches | 2 |
| PPO Clip | 0.2 |
| Clipped value loss | No |
| Per mini-batch advantage normalization | No |
| $\gamma$ | 0.99 |
| GAE-$\lambda$ (Schulman et al., 2016) | 0.95 |
| Learning rate | $5.0 \times 10^{-4}$ Depth, $2.5 \times 10^{-4}$ RGB |
| Learning rate scaling | $\sqrt{\frac{B}{B_{\text{base}}}}$ |
| $B_{\text{base}}$ | 256 |
| Max gradient norm | 1.0 |
| Weight decay | 0.01 |
| Lamb $\rho$ | 0.01 |
| Per GPU parameters | |
| Number of unique scenes ($K$) | 4 |
| Simulation batch size/Number of Environments ($N$) | 128 |
| Rollout length ($L$) | 32 |

Table A4: Hyper-parameters used for BPS training on 8 GPUs.

**CPU details**. Each GPU configuration also uses different CPU configurations based on hardware access. Tesla V100 benchmarking was done with 2x Intel Xeon E5-2698 v4 (a DGX-1 station). RTX 2080 TI benchmarking was done with 2x Intel Xeon Gold 6226. RTX 3090 benchmarking was done with with 1x Intel i7-5820k. On all CPUs, we disable Hardware P-State (HWP) (where applicable) and put software P-State in performance mode. Our CPU load on simulation worker cores is inherently sporadic and we find that certain CPUs are unable to change clock frequencies fast enough to not incur a considerable performance penalty when allowed to enter a power saving state.

| Sensor | System | CNN | Resolution | | Tesla V100 | | RTX 2080Ti | | RTX 3090 |
|---|---|---|---|---|---|---|---|---|---|
| | | | | | 1 GPU | 8 GPUs | 1 GPU | 8 GPUs | 1 GPU |
| Depth | BPS | SE-ResNet9 | 64 | PPO Epochs | | | 1 | | |
| | | | | Rollout length ($L$) | | | 32 | | |
| | | | | Number of Environments ($N$) | 1024 | 128 | 512 | 128 | 1024 |
| | BPS | SE-ResNet9 | 128 | PPO Epochs | | | 1 | | |
| | | | | Rollout length ($L$) | | | 32 | | |
| | | | | Number of Environments ($N$) | 512 | 128 | 128 | 128 | 512 |
| | BPS-R50 | ResNet50 | 64 | PPO Epochs | | | 1 | | |
| | | | | Rollout length ($L$) | | | 32 | | |
| | | | | Number of Environments ($N$) | 512 | 128 | 256 | 128 | 512 |
| | BPS-R50 | ResNet50 | 128 | PPO Epochs | | | 1 | | |
| | | | | Rollout length ($L$) | | | 32 | | |
| | | | | Number of Environments ($N$) | 256 | 128 | 64 | 64 | 128 |
| | WIJMANS++ | SE-ResNet9 | 64 | PPO Epochs | | | 1 | | |
| | | | | Rollout length ($L$) | | | 32 | | |
| | | | | Number of Environments ($N$) | | | 64 | | |
| | WIJMANS20 | ResNet50 | 128 | PPO Epochs | | | 2 | | |
| | | | | Rollout length ($L$) | | | 128 | | |
| | | | | Number of Environments ($N$) | | | 4 | | |
| RGB | BPS | SE-ResNet9 | 64 | PPO Epochs | | | 1 | | |
| | | | | Rollout length ($L$) | | | 32 | | |
| | | | | Number of Environments ($N$) | 512 | 128 | 128 | 128 | 256 |
| | BPS | SE-ResNet9 | 128 | PPO Epochs | | | 1 | | |
| | | | | Rollout length ($L$) | | | 32 | | |
| | | | | Number of Environments ($N$) | 256 | 128 | 64* | 64* | 256 |
| | BPS-R50 | ResNet50 | 64 | PPO Epochs | | | 1 | | |
| | | | | Rollout length ($L$) | | | 32 | | |
| | | | | Number of Environments ($N$) | 256 | 128 | 64 | 64 | 256 |
| | BPS-R50 | ResNet50 | 128 | PPO Epochs | | | 1 | | |
| | | | | Rollout length ($L$) | | | 32 | | |
| | | | | Number of Environments ($N$) | 128 | 128 | 32* | 32* | 64 |
| | WIJMANS++ | SE-ResNet9 | 64 | PPO Epochs | | | 1 | | |
| | | | | Rollout length ($L$) | | | 32 | | |
| | | | | Number of Environments ($N$) | 20 | 20 | 6 | 6 | 16 |
| | WIJMANS20 | ResNet50 | 128 | PPO Epochs | | | 2 | | |
| | | | | Rollout length ($L$) | | | 128 | | |
| | | | | Number of Environments ($N$) | | | 4 | | |

Table A5: System configuration parameters for Table 1. * indicates 4 mini batches per epoch instead of 2.

