# OpenReview forum: "Large Batch Simulation for Deep Reinforcement Learning"
_ICLR.cc/2021/Conference — ICLR 2021 Poster_

### Official Review · AnonReviewer3 · 2020-10-16
**Well written paper on improving resource efficiency by batched environment simulation and rendering**

**Rating:** 7
**Confidence:** 4

**Review:**

The paper achieves a significant speed-up for the PointGoal Navigation task by,
- Batched simulation of  environments (N > #CPUs)
- Batched rendering of multiple environments with a single rendering request, effectively drawing the environments as one big frame
- Reusing scene assets between environments being rendered in a way that doesn't hurt sample diversity but allows more environments to be rendered at the same time
- Using multiple accelerators
- Shows that comparable results can be obtained faster

The paper is well-written and clear. Similar ideas have been tried out but the combination of the ideas is well done and with a large efficiency improvement. One downside is that the setup is not easily applicable in general.

Pros:
- Well written and clear.
- Large efficiency gain with the same amount of resources. Allowed faster research iteration.
- Analysis on runtime (4.5) is clear and insightful.

Cons:
- Not easily transferable to other environments.
- The paper would benefit from plots showing agent performance over time and plots that shows how sample efficiency hurts with scale and increased batch size. As well as stability across time.

Comments:
- In abstract, it is mentioned one GPU setup achieves 19k FPS and 8 GPU setup 72k FPS. These two numbers are with different types of GPUs which makes it look like the speed-up is only 4x going to 8 GPUs when it's really 6x (72000/12900) and possibly more if batch size per GPU is kept the same.
- Wrt. section 3.3, it's mentioned that an efficient network is used (Resnet + 2xLSTM) compared to a less efficient CNN + LSTM network and that it has lower capacity. I assume most of the capacity in the CNN+LSTM network is from initial layers with the large input and a FCN before the LSTM?  What effects does the changes have? e.g. does 256^2 -> 64^2 alone hurt in any way? I suspect the Resnet makes up for some of it?

---

> ### Author Response · Authors · 2020-11-25
> **Response to Review #3 (Part 1/2)**
>
> We thank the reviewer for their detailed review and for clearly outlining the conceptual contributions of our submission that lead to high simulation performance: batch simulation, batch rendering, and 3D asset sharing without hurting sample efficiency. We appreciate the reviewer’s comment that this “combination of the ideas is well done” and recognition of our “large efficiency improvement.” We have answered the reviewer’s comments and questions in detail below:
>
> > **The setup is not easily applicable in general -- not easily transferable to other environments.**
>
> The generality of our work and the effort required to bring these performance improvements to new environments was a common concern across multiple reviewers, so we have addressed it in our top-level comment here: https://openreview.net/forum?id=cP5IcoAkfKa&noteId=ghsVTvuKaow.
>
> Additionally, we have added two brief experiments on new tasks using a new dataset (AI2-Thor) to demonstrate batch rendering and simulation on tasks other than Pointgoal navigation, as described in Appendix A.1.
>
> > **Need plots showing agent performance and stability over time.**
>
> We have added a new graph, Figure 3, that shows validation set SPL versus wall-clock time for training a RGB camera driven agent on a RTX 3090 for BPS, Baseline++ and Baseline, as well as discussion of these results in Section 4.4. These results demonstrate that training with BPS is stable, in fact BPS’s validation score is more stable than the other systems due to its large training batches, as well as the fact that Baseline results in the graph are dominated by the early, unstable, phase of training due to the large performance gap between systems. In addition to a smooth validation score throughout training, BPS also converges to within 1% SPL of its peak performance within just over 20 hours of training, at approximately 1 billion samples of experience -- less than half the 2.5 billion samples used for convergence for consistency throughout the paper.
>
> > **Need data showing how sample efficiency is affected by batch size.**
>
> We added Figure 4 in Section 4.4, which shows SPL versus wall-clock time for a depth-driven agent across a range of batch sizes, both smaller and larger than the default settings described in Section 4.1. Additional data showing SPL versus number of training samples for this experiment is also provided in Figure 7. Compared to the standard batch size across an eight GPU run of 1024 rollouts, a batch size of 256 does improve sample efficiency; however, the greater than 2x performance improvement from the larger batch size makes it much more practical in terms of wall-clock training time to reach a given target SPL. This is consistent with observations in prior and concurrent work (Espeholt et al 2020, Dalton et al 2020).
>
> > **The abstract mentions single GPU and eight GPU results on different GPU modes… which makes it look like the speed-up is only 4x going to eight GPUs when it’s really 6x.**
>
> The reviewer’s analysis is correct. We presented our best results for both single and eight GPU configurations in the abstract to match our philosophy of maximizing performance on available hardware rather than focusing on scalability to the detriment of absolute performance. Unfortunately, we do not have access to eight RTX 3090s to make this comparison on the same GPU model.
>
>  > **Section 3.3 mentions an efficient network (Resnet + 2x LSTM) is used compared to a less efficient CNN + LSTM network… What effects do the architecture changes have? I suspect the Resnet makes up for some of it?**
>
> We want to clarify that both the Baseline and our policy DNN use a ResNet-based visual encoder (we have updated the text to clarify the prior work’s visual encoder architecture in Section 3.3). The baseline uses a ResNet50-based visual encoder, while our optimized policy DNN uses a ResNet9-based visual encoder (as described in Section 3.3). Unfortunately, we lack the computational resources to train a ResNet50-based policy to 2.5 billion frames with BPS within the rebuttal deadline. Our existing results in Table 2 demonstrate the performance of the ResNet50-based policy using the baseline system (as published in DD-PPO, ICLR 2020) versus BPS training a ResNet9-based architecture.

---

> > ### Author Response · Authors · 2020-11-25
> > **Response to Review #3 (Part 2/2)**
> >
> > > **Does an input resolution reduction from 256x256 to 64x64 hurt task performance?**
> >
> > Note that the baseline architecture performs an initial 2x2 downsampling operation to reduce the effective input resolution to 128x128; therefore BPS’s ResNet9 64x64 input resolution is only a factor of two reduction in each dimension (we have added text clarifying the behavior of the baseline in Sections 3.3 and 4.1).
> >
> > The input resolution reduction does not hurt task performance. To show this, we have added a new row to Table 2, which runs BPS’s ResNet9 visual encoder with an input resolution of 128x128, using downsampled 256x256 frames from the renderer (to match Baseline’s behavior). When using this higher resolution input, the ResNet9-based policy used by BPS experiences a slight reduction in validation set SPL (within the margin of error) relative to BPS at 64x64 pixels; we do not have access to the test set in the rebuttal time-frame, but these numbers will be added later. Additionally, end-to-end performance of BPS with this resolution change (along with results using the baseline’s policy DNN), have been added to Table 1 and show that BPS’s end-to-end performance is only reduced by ~2x after increasing the resolution from 64x64 to 128x128 (allowing BPS to therefore maintain a substantial lead over the baselines even at a higher resolution).

---

### Official Review · AnonReviewer2 · 2020-10-29
**Large Batch Simulation for Deep Reinforcement Learning**

**Rating:** 7
**Confidence:** 3

**Review:**

This paper proposed a novel RL simulator design that executes large batches of simulation simultaneously. The core innovation lies in an efficient GPU-based batch rendering and an improved training procedure adapted for large mini-batch optimization. Compared to strong baselines, the method achieves a significant boost in efficiency with slightly reduced performance.

Pros:
* The paper tackles a very critical speed bottleneck in practice for RL training in a simulation environment.
* Very significant speed improvement
* The paper is well-written and presented

Cons:
* The paper misses some important ablations to showcase each technical component's contribution to the final speedup.
* Evaluation is only conducted on PointNav task; many technical design choices seem to be tailored for such a task, e.g., very low-resolution images, etc.
* It seems the choices of DNN architecture, policy optimization, and batch rendering are highly coupled. It's not clear to me if it is possible to use batch-renderer off-the-shelf to accelerate the majority of RL algorithms' training.


I found some ablation is lacking, making it unclear how much each part contributes to the final acceleration / reduced performance. For instance, it is unclear to me how much the frame rate increases by changing the resolution from 256x256 to 64x64 and how much performance drop it brings. Further, how much acceleration of BPS is due to the simplified DNN architecture choice (shallow layers, SE blocks, SpaceToDepth, etc.)

In many RL tasks, it might be necessary to render high-resolution images. Thus it's essential to know if the proposed method is suitable for such a job. Could you provide an acceleration-vs-resolution curve of the proposed batch renderer?

It seems the simulation takes little time in breakdown compared against rendering and network inference/training. Is this framework suitable for heavy non-rendering simulation jobs, e.g., if actor simulation or physical dynamics simulation is involved?

Could you justify whether this proposed BPS is also suitable for multi-agent RL training, in which multiple agents might share one environment?

The author should elaborate a bit more on why large-batch training harms sample efficiency.

Could you discuss the batch rendering acceleration compared to other graphics techniques to improve rendering efficiency, such as level-of-details (LOD), mesh simplification, DLSS, etc? And if the proposed batch rendering could be used together with these techniques?

-------------------------------------------------------- Post Rebuttal --------------------------------------------------------------------------------------------

The author did a great job in terms of addressing most of my concerns and answer all the questions. I also like the updated paper. The update reflects most major concerns from the reviewers. Thus I would like to raise my score to 7 and would like to champion this paper. I do think the paper would potentially provide great value to the community not only due to its open-source effort but also as a general approach to improve simulator efficiency for RL, despite it's not "novel" methodology-wise.

---

> ### Author Response · Authors · 2020-11-25
> **Response to Review #2 (Part 1/2)**
>
> We thank the reviewer for their detailed review and positive comments regarding our “novel RL simulator design” as well as the “very significant speed improvement” that we’ve achieved while addressing a “critical speed bottleneck in practice for RL training.” Additionally, we appreciate their thoughtful comments and questions, which are answered in detail below:
>
> > **DNN architecture, policy optimization, and batch rendering seem highly coupled. Can batch rendering be used off-the-shelf to accelerate the majority of RL training?**
>
> Batch rendering can accelerate 3D rendering regardless of DNN architecture, rendering resolution, or policy optimization algorithm. Batch rendering is the idea of using knowledge of multiple rendering operations to render multiple environments more efficiently: e.g., by amortizing memory resources and CPU/GPU communication work across operations. This core idea is only dependent on multiple environments being simultaneously available for simulation and rendering. Fortunately, prior work (e.g. Sample Factory, SEED RL) has demonstrated that *batched DNN inference* can be implemented with a range of RL algorithms, which naturally complements batch rendering’s need for a synchronous batch of simulation states.
>
> Speedup from adding batched rendering to off-the-shelf RL systems will depend on whether or not rendering is the bottleneck.  When rendering is the bottleneck, batch rendering can yield considerable speedups without any further changes. If other system components are the bottleneck (such as the DNN inference/training), these components must also be optimized to see significant end-to-end performance gains; this is exactly the motivation for the DNN architecture optimizations we present in this paper.
>
> To show how end-to-end performance gains of batch rendering/simulation change with different DNN architectures, we provide new results in Table 1 that show BPS’s performance when used with the same ResNet50 DNN architecture and image resolution as Baseline, in addition to results using the optimized ResNet9-based architecture. On an RTX 3090, BPS achieves a 10x speedup over Baseline while using the same DNN architecture (at the same input resolution). Additionally, Table 3 has been added with a breakdown of BPS’s runtime when running with the ResNet50 based architecture (as well as breakdowns of the baselines).
>
> > **High-resolution images may be necessary for broader RL applications. How does performance change for the batch renderer implementation in BPS at different resolutions?**
>
> Batch rendering is applicable at any resolution. We have added Figure 6 in Appendix A.2 that shows performance (FPS) of the standalone batch renderer for different resolutions and batch sizes. In the geometrically complex Gibson scenes used for this benchmarking, rendering performance is limited by triangle processing (Section 3.2), so rendering higher resolution output has marginal additional cost. For example, when rendering with a batch size of 512, moving from 32x32 pixel frames to 512x512 pixel frames results in a performance decrease from 48000 FPS to 23000 FPS: only a 2x reduction in performance despite the 256x increase in pixel count.
>
> > **How does each contribution affect the final speedup achieved? Need an ablation study between DNN input resolution, simplified DNN architecture and batch simulation / rendering.**
>
> Baseline++ uses the same policy DNN (and input resolution) as BPS so it ablates the batch simulator and renderer. Table 1 shows that for depth sensor agents, BPS achieves a 4.5x to 7.8x speedup over Baseline++, and for RGB sensor agents, BPS achieves a 6x to 13x speedup over Baseline++.
>
> We added new experiments to Table 1 that demonstrate the performance of BPS with the same DNN architecture and input resolution as Baseline to ablate the effect of our ResNet9 DNN architecture. On an RTX 3090 (for both depth and RGB agents), BPS (with Baseline’s ResNet50 and input resolution) still outperforms Baseline by over 10x. Additional ablations are also included showing performance of both the ResNet9 DNN and ResNet50 DNN at different resolutions.
>
> We note that these ablations do not entirely isolate the performance benefit provided by each contribution because GPU memory usage entangles the performance of each component in the system. For example, a larger policy DNN increases the memory usage of policy optimization, which may limit the maximum simulation/inference batch size that can fit in GPU memory, impacting performance of all components of the system. Nevertheless, these results show that BPS still performs much faster (10x on an RTX 3090) than the baseline even with the same policy DNN and input resolution.

---

> > ### Author Response · Authors · 2020-11-25
> > **Response to Review #2 (Part 2/2)**
> >
> > > **How much is task performance affected by the reduction from 256x256 input resolution to 64x64 input resolution for the policy DNN?**
> >
> > Note that the baseline architecture performs an initial 2x2 downsampling operation to reduce the effective input resolution to 128x128; therefore, BPS’s ResNet9 64x64 input resolution is only a factor of two reduction in each dimension (we have added text clarifying the behavior of the baseline in Sections 3.3 and 4.1).
> >
> > The input resolution reduction does not hurt task performance. To show this, we have added a new row to Table 2, which runs BPS’s ResNet9 visual encoder with an input resolution of 128x128, using downsampled 256x256 frames from the renderer (to match Baseline’s behavior). When using this higher resolution input, the ResNet9 based policy used by BPS experiences a slight reduction in validation set SPL (within the margin of error) relative to BPS at 64x64 pixels; we do not have access to the test set in the rebuttal time-frame, but these numbers will be added later.
> >
> > > **Evaluation is only on Pointgoal navigation.**
> >
> > We have added two brief experiments on new tasks using a new dataset (AI2-Thor) to demonstrate batch rendering and simulation on tasks other than Pointgoal navigation, as described in Appendix A.1. Additionally, generality was a common concern across multiple reviewers, so we have addressed it in our top-level comment here: https://openreview.net/forum?id=cP5IcoAkfKa&noteId=ghsVTvuKaow.
> >
> > > **Is this framework suitable for heavy non-rendering simulation jobs, e.g., if actor simulation or physical dynamics simulation is involved?**
> >
> > In this paper, we showed that batch simulation reduces the computational cost and memory footprint of the non-rendering simulation aspects of training PointGoal navigation agents. We believe that future simulators (e.g those that simulate more complex actor behavior or more complex physics) would also benefit from the ability to optimize over multiple environment simulation requests at once to perform optimizations such as reducing synchronization overheads and increasing available workload parallelism, especially when moving a larger portion of the simulation workload to accelerators such as GPUs.
> >
> > > **What is the relationship between batch rendering and other graphics techniques to improve rendering efficiency?**
> >
> > Optimizations enabled by batch rendering are orthogonal and complementary to other advanced rendering techniques such as: level-of-detail techniques, mesh simplification, and super-resolution techniques like DLSS. We note that our rendering implementation makes use of frustum culling as one level-of-detail technique (batch rendering makes it easier to efficiently implement frustum culling since the culling of triangles from one scene is pipelined with the rendering of triangles for another scene in the batch). The other techniques listed above would likely further improve performance. However, as shown in Section 4.5, the primary bottleneck to end-to-end performance in BPS is DNN inference/training. Therefore, further performance optimization of those components would have more impact on end-to-end performance.
> >
> > > **Is batch simulation suitable for multi-agent RL training?**
> >
> > Yes. Multi-agent RL training complements batch simulation (in particular, batch rendering) by producing multiple rendering and simulation operations within the same environment. Multi-agent training naturally provides an increased number of simultaneous simulation / rendering requests without needing to increase the number of simultaneous rollouts. Additionally, multi-agent training could be used to provide further optimization opportunities in a batch rendering system: e.g., amortizing complex lighting computations across rendering requests involving the same scene.
> >
> > > **Why does large batch training harm sample efficiency?**
> >
> > The effect of large mini-batch optimization on learning performance (be it generalization performance or sample efficiency) is a complex question that is an active area of research in the deep learning community and far beyond the scope of this submission. Refer to Dinh et al; 2017 (ICML), Keskar et al; 2017 (ICLR), Goldblum et al; 2020 (ICLR), and Johnson et al; 2020 (ICML) for some of the current work in this area.
> >
> > To provide empirical evidence for the relationship between batch size and sample efficiency in our results, we added Figure 4 (and text in Section 4.4), which shows SPL versus time for a depth-driven agent across a range of batch sizes, both smaller and larger than the default settings described in Section 4.1. Additional data showing SPL versus number of training samples for this experiment is also provided in Figure 7. With the Lamb optimizer, the difference in final converged SPL is less than 1% between different batch sizes, with the smallest batch size achieving the highest SPL (at the cost of worse performance), and the largest batch size converging to the lowest SPL.

---

### Official Review · AnonReviewer4 · 2020-10-29
**A useful (engineering) contribution to the Deep RL community, that lacks some comparisons and details.**

**Rating:** 5
**Confidence:** 3

**Review:**

This work extends the Habitat simulator to perform large batch training. Although this is mainly an engineering contribution, the approach is useful for the community. However I have some questions:


1. You do not appear to specify the CPU memory requirements of this approach, I believe each Habitat environment instance requires a significant amount of memory, how many independant instances can be loaded in paralel with this approach and what is the memory requirement? What is the value of K that is discussed on the last paragraph of page 4?

2. How significant is the use the Lamb optimizer, can you include an ablation of this, do the other baselines use this optimizer?
3. In table 2 you do not show the performance of the Baseline++ network architecture
4. In table 2 some entries have standard error and some do not, if these results are now available please can they be included.
5. Would it be possible to update figure 3 with comparisions of the baseline methods.

---

> ### Author Response · Authors · 2020-11-25
> **Response to Review #4**
>
> We thank the reviewer for recognizing that the approach presented in our submission is “useful for the community.” We would like to emphasize our belief that this submission represents more than a simple engineering contribution; instead, we introduce a novel architecture for high-performance simulation in visually complex environments that is made up of several intellectual contributions: the core idea of batch simulation, batch rendering of complex environments, and a number of contributions to the DNN workload to maintain and improve training efficiency. We have answered the reviewer’s comments and questions in detail below:
>
> > **This work extends the Habitat simulator to perform large batch training.**
>
> First, we would like to clarify the misconception that this work extends an existing simulator. Our implementation of batch simulation -- the BPS simulator -- is a standalone simulator, not an extension of Habitat (note that the code provided in the supplemental material leverages a modified version of Habitat’s implementation of DD-PPO (our training algorithm), and Habitat’s simulator is used for validation to enable clean comparison with prior work, which may be the source of confusion on this point). Please refer to our top-level comment here: https://openreview.net/forum?id=cP5IcoAkfKa&noteId=ghsVTvuKaow, for a more detailed discussion of the intended goal of the BPS simulator.
>
> > **Each Habitat environment instance requires a significant amount of memory… What is the CPU memory requirement?**
>
> Batch simulation allows a single simulator instance to simulate and render hundreds (or thousands) of environments in parallel, which significantly reduces usage of both CPU and GPU memory compared to the strategy of duplicating simulator instances used by prior work (including Habitat). Concretely, our implementation of batch simulation, BPS, uses about 5 GB of CPU memory while simulating and training 1024 environments on a single GPU. Conversely, Baseline++ (which uses the Habitat Simulator) consumes 50 - 60 GB of memory while simulating 64 environments with separate simulator instances.
>
> > **What is the value of K that is discussed on the last paragraph of page 4?**
>
> K refers to the number of unique 3D scene assets that are used in a batch of N simulated/rendered environments. We define its value in the original submission in Section 4.1 (“Determining Batch Size”). We use one unique scene asset per 32 environments in the batch. (So the per-GPU batch size of 128 in our eight-GPU experiments would translate to K = 4 per GPU). We agree that the concrete value of this parameter could have been emphasized more clearly and have added wording to Section 4.1 giving an explicit example of K’s value for BPS. For the baselines leveraging the Habitat simulator, K is equal to the number of simulator instances, as no asset sharing is possible (values listed in Section 4.1, “Determining Batch Size”).
>
> > **How significant is the use of the Lamb optimizer ... Do the baselines use Lamb?**
>
> The Lamb optimizer increases final SPL for Pointgoal navigation by 1.5% for depth-sensor driven agents in the first 1 billion samples of training and by 0.6% after 2.5 billion samples of training. We have added a new graph demonstrating this improvement: Figure 8 in Appendix A.3. The baselines do not use Lamb, as our use of Lamb was motivated by the large batches used to maximize the performance of batch simulation.
>
> > **Table 2 does not show the Baseline++ network architecture.**
>
> Baseline++ and BPS use the same policy DNN (Section 4.1, “Baseline++”). Therefore, the BPS results in Table 2 do provide the performance of Baseline++’s network architecture (as it is the same as BPS). We did not train the same architecture using the Baseline++ system due to the amount of time required (>2 weeks on 8 V100s, over $10k at current AWS pricing, for RGB).
>
> > **Missing some standard errors in Table 2.**
>
> The test set for Gibson is behind an evaluation server (https://eval.ai/web/challenges/challenge-page/254/overview). Therefore we are unable to get standard error for the test set. Baseline numbers are taken from the published DD-PPO results, which did not provide standard error for RGB on the validation set.
>
> > **Can the runtime breakdown (figure 3) be updated with comparisons of baseline methods?**
>
> Due to the significant performance difference between BPS and the baselines, including the baselines in Figure 3 would render the BPS results illegible. We have added Table 3 showing the runtime breakdowns for the baselines (as well as BPS with a larger policy DNN for comparison) to sidestep the limitations of the figure format.

---

### Official Review · AnonReviewer1 · 2020-10-29
**A Nice Enginnering Work**

**Rating:** 6
**Confidence:** 2

**Review:**

The paper proposed a high-performance reinforcement learning system for training agents in 3D simulated environments.
The system uses batching to significantly increases both simulation time and training time.
It also proposes tricks for large mini-batch policy optimization.

The experiments do evaluate several important aspects of the proposed method. However, I want to see how generalizable the system design is. The current paper only evaluates the system for one environment and one algorithm, which is a major weak point.

---

> ### Author Response · Authors · 2020-11-25
> **Response to Review #1**
>
> We thank the reviewer for their comments and review. We would like to emphasize our belief that this submission represents more than a simple engineering contribution; instead, we introduce a novel architecture for high-performance simulation in visually complex environments, which is made up of several intellectual contributions: the core idea of batch simulation, batch rendering of complex environments, and a number of contributions to the DNN workload to maintain and improve training efficiency.
>
> > **Generalizability of the system design is not demonstrated: the current paper only evaluates the system for one environment and one algorithm.**
>
> The primary goal of this paper is to demonstrate how batch 3D rendering and simulation can accelerate RL training in complex 3D environments, regardless of DNN architecture, rendering resolution, or policy optimization algorithm. Batch rendering is the idea of using knowledge of multiple rendering operations to render multiple environments more efficiently: e.g., by amortizing memory resources and CPU/GPU communication work across operations. This core idea depends only on multiple environments being simultaneously available for simulation and rendering in the RL system.
>
> The Pointgoal navigation task and two detailed datasets (Gibson and Matterport3D) we train on in this paper serve as examples for demonstrating the significant efficiency gains made possible by our contributions. While batch simulation is certainly applicable to other RL tasks, our goal is not to introduce a fully-featured RL agent training platform for general use. Instead, we hope our ideas will be incorporated into existing RL platforms to significantly improve their performance. Since submission, we have met with the developers of existing RL simulators and have heard that they are excited to incorporate the ideas presented in our paper in their systems. As generalizability was a common concern across multiple reviewers, we have addressed it in our top-level comment here: https://openreview.net/forum?id=cP5IcoAkfKa&noteId=ghsVTvuKaow.
>
> Additionally, to address this concern concretely, we have added two brief experiments on new tasks using a new dataset (AI2-Thor) to demonstrate batch simulation and rendering on tasks other than PointGoal navigation, as described in Appendix A.1.

---

### Official Review · AnonReviewer5 · 2020-11-09
**limited novelty and impact**

**Rating:** 4
**Confidence:** 4

**Review:**

This paper shows that batch simulation can accelerate reinforcement learning in 3D environments. Batch simulation accepts and executes large batches of simulation requests at the same time on one accelerator. The authors demonstrate that this technique can substantially speed up the processing and achieve ~100x speed up in convergence. They also propose minor-changes to DD-PPO to speed up the convergence even further. The authors also included the code which is always appreciated.

I believe the problem that this paper is addressing is quite important. Long training time of RL agents is a big issue which makes evaluation of the current methods harder and, as authors mentioned in the introduction, unaccessible to many people due to limited number of people who have access to large clusters of computation.

However, I find the paper limited in both novelty and impact. The idea of "batch" simulation is not new and it has been shown before, not only for complex environments (check Sample Factory from ICML 2020 for an async version) but also for simple ones such as atari when ported to the GPU (check NVlabs CuLE from NeurIPS 2020). Also given the limited number of the environments that the paper covers and (probably) the complexity of adding new ones, it has limited impact.

Overall, I find this to be a great technical paper. The authors achieved remarkable convergence speed for PointGoal navigation problem and I believe that upon releasing the code some researchers will benefit from it. However, I find it lacking in novelty and future impact as a scientific paper for a top machine learning conference.

---

> ### Author Response · Authors · 2020-11-25
> **Response to Review #5**
>
> > **The idea of "batch" simulation is not new and it has been shown before**
>
> The idea of batch simulation and rendering is for the implementation of these components to perform processing, memory footprint, and command/data communication optimizations that are possible only when a large number of requests are exposed to the renderer (or simulator) at once (Section 3, paragraph 3).
>
> Under this definition, Sample Factory (Petrenko et al, ICML 2020) *does not use batch simulation*.  Instead, each actor worker performs simulation for its environments independently and serially (we have confirmed this via personal correspondence with the authors and examination of the published code). The actor workers are independent separate processes that do not share rendering assets (even between the environments on the same worker) and each generates rendering operations for only a single image at a time. These limitations would prevent Sample Factory from efficiently simulating visually complex environments that require GPU acceleration and are exactly the problems that are solved by our introduction of batched rendering and simulation (as described in Section 3.2), which enable the two order of magnitude speedups we present in this paper.
>
> CuLE (Dalton et al, NeurIPS 2020) is an example of batched simulation/rendering. We were unaware of CuLE at the time of submission -- thank you for the reference. This work is concurrent work as per ICLR’s policies (https://iclr.cc/Conferences/2021/ReviewerGuide#faq) as it is yet to be published (and was only accepted for publication a week before the ICLR deadline). We have update our related work section to include the following:
>
>
> Concurrent with our work, Dalton et al. (2020) propose CuLE, a batched GPU-accelerated reimplementation of the Atari Learning Environment (ALE).  Compared to CuLE, our focus is on implementing a high-performance batched render for complex 3D environments, which involves optimizations (GPU command buffer optimizations, memory asset sharing, async environment paging) not addressed by CuLE because of the simplicity of rendering in Atari-like applications. Additionally, we improve the relationship between batch simulation and sample efficiency by showing that the reduction in sample efficiency due to large mini-batch optimization can be alleviated via large batch optimization techniques from the supervised learning literature.
>
> > **Also given the limited number of the environments that the paper covers and (probably) the complexity of adding new ones, it has limited impact.**
>
> As a direct response to the comment that this paper has limited impact, we would like to emphasize that the novel ideas presented in this paper, not necessarily the code artifact we present (the BPS system), have the largest potential for impact across a broad range of present and future RL systems. We address this concern in more detail in our top-level comment here: https://openreview.net/forum?id=cP5IcoAkfKa&noteId=ghsVTvuKaow.
>
> Additionally, we have added two brief experiments on new tasks using a new dataset (AI2-Thor) to demonstrate batch simulation and rendering on tasks other than PointGoal navigation, as described in Appendix A.1.

---

### Author Response · Authors · 2020-11-25
**Generality and Impact of this Submission**

Training embodied agents in complex simulated 3D environments is central to progress in RL, and a major concern of many widely used training platforms (e.g. AI Habitat, AI2-THOR, ThreeDWorld, Isaac Sim, Gibson, Unity ML Agents, Carla) is the efficiency of rendering and simulating virtual environments.  In this paper, we demonstrate how to architect a 3D rendering system “from-the-ground-up” for RL-workloads. Specifically, we demonstrate that a rendering architecture designed for rendering many scenes and many scene viewpoints *en masse* (“batch rendering”) enables new optimizations (GPU/CPU command stream optimization, high GPU utilization, scene asset sharing for reduced footprint, computation-computation overlap). These optimizations are not specific to any dataset or task. As a result of this design, rendering complex 3D scenes from Gibson, Matterport3D, and (in the submitted revision) AI2-Thor is *no longer the dominant cost* (neither in terms of compute nor memory) during training, even after inference/training is optimized to use small/computationally efficient DNN architectures.  Additionally, for the specific case of Pointgoal navigation, *simulating* many environments *en masse* (“batch simulation”) significantly reduces the CPU cost and memory footprint of simulation.  As a result, our end-to-end training performance is 10 times faster than prior work using an *identical* agent DNN architecture (and input resolution). Optimizing the DNN architecture to keep up with the faster rendering and simulation components yields an additional 10x performance improvement. The cumulative result of these optimizations is that experiments that previously took three days on a 64-GPU cluster can now complete in one day on a single GPU.

Our goal is not to provide the community a new software platform for ML training, as many excellent platforms already exist. However, we hope that existing platforms that wish to realize significantly higher performance in complex 3D environments consider adopting our proposed system architecture and ideas. We believe that using batched rendering and simulation techniques to make efficient use of CPU/GPUs, rather than designing systems that scale out to many machines to overcome their single-node inefficiencies, is a more promising way to build future systems. This will allow RL researchers to benefit from fast turn around using modest computing hardware that is widely available.

We will release the BPS renderer and end-to-end Pointgoal task implementation open source to serve as a reference implementation that will facilitate integration of our ideas into current and future RL systems.

We have updated the abstract to more clearly reflect how the main contribution of our work is a set of system architectural decisions and implementation techniques (that stand to benefit many existing RL training systems) rather than a new software platform designed for use by RL researchers.

---

### Decision · Program_Chairs · 2021-01-07
**Final Decision**

**Decision:**

Accept (Poster)

**Comment:**

This paper proposes techniques to lower the barrier to run large scale simulations under resource (compute) constraints. The key idea is to do batch simulation and policy learning on 1 or more GPUs without sacrificing the fps rate for rendering (~20k fps on 1GPU). The proposed setup and methods are evaluated on the point navgiation tasks on two environments namely Gibson and Matterport3D. One of the key ideas for rendering is to render a big tile of images for separate instantiations of the environment in parallel. This gives big speeds up to rendering and policy optimization.

${\bf Pros}$:
1. Large number of FPS with smaller compute budget. Large scale Deep  RL research has been difficult to democratize due to the need for big compute budgets. Although this paper is more heavy on the engineering side, I think it can greatly accelerate research and therefore seems like a good fit for the ICLR community to consider.

2. The paper and proposed steps are clearly written and justified

${\bf Cons}$:
1. This method is limited to environments where the observation space follows a particular structure. This is perhaps the biggest limitation of this approach but the underlying assumptions are reasonable and quite a few realistic environments will fall into this category.


During the rebuttal and discussion period, R2 raised concerns about ablations but was satisfied with author's response. R5 raised concerns about other prior work - CuLE (Dalton et al, NeurIPS 2020). However, this paper is concurrent work and does not tackle 3D simulation rendering as done in this paper. For these reasons I believe the paper does not have any big red flags or pending concerns.